# Lifestyle Factors Associated with Metabolic Syndrome in Urban Cambodia

**DOI:** 10.3390/ijerph191710481

**Published:** 2022-08-23

**Authors:** Miharu Tamaoki, Ikumi Honda, Keisuke Nakanishi, Maki Nakajima, Sophathya Cheam, Manabu Okawada, Hisataka Sakakibara

**Affiliations:** 1Department of Integrated Health Sciences, Graduate School of Medicine, Nagoya University, Nagoya 461-8673, Japan; 2Department of Pediatric, Sunrise Japan Hospital Phnom Penh, Phnom Penh 121001, Cambodia; 3School of Nursing, Ichinomiya Kenshin College, Ichinomiya 491-0063, Japan

**Keywords:** metabolic syndrome, Cambodia, lifestyle factor, non-communicable diseases

## Abstract

This study aimed to identify lifestyle factors associated with metabolic syndrome (MetS) in urban Cambodia. In this cross-sectional study, we used existing health checkup data from a private hospital in Phnom Penh, Cambodia. The participants comprised 5459 Cambodians aged ≥20 years who underwent health checkups between 2017 and 2019. The harmonized diagnostic definition was used as the MetS criteria. The prevalence of MetS was 56.6% overall, 60.4% in men and 52.6% in women. The lifestyle factor significantly associated with MetS in both sexes were “eating quicker than others”, (men: odds ratio [OR]= 2.25, 95% confidence interval [CI] = 1.68–3.03, women: OR = 1.92, 95%CI = 1.41–2.60), “walking faster than others”, (men: OR = 0.78, 95% CI = 0.67–0.92, women: OR = 0.75, 95% CI = 0.62–0.89) and “drinking alcohol” (men: OR = 1.33, 95% CI = 1.10–1.61, women: OR = 1.33, 95% CI = 1.09–1.62). Other significant associations with MetS for men was “eating speed is normal”, (OR = 1.73, 95%CI = 1.30–2.31), and, for women, “eating food after dinner at least 3 days a week”, (OR = 1.25, 95%CI = 1.01–1.55), “skipping breakfast at least 3 days a week”, (OR = 0.83, 95%CI = 0.69–0.99) and “getting enough rest from sleep” (OR = 1.19, 95% CI = 1.01–1.42) were significantly associated with MetS. Lifestyle interventions through health education and guidance may be effective in preventing MetS in Cambodia.

## 1. Introduction

Cambodia, a lower-middle-income country, has experienced a marked increase in the prevalence of non-communicable diseases (NCDs) such as cardiovascular disease, stroke and type 2 diabetes mellitus [1,2]. NCDs account for 77% of global deaths in low- and middle-income countries [3], and the causes of death within Cambodia have changed dramatically from communicable diseases to NCDs over the past few decades. In 2019, 68% of deaths were from NCDs [4]. One factor contributing to the increase in NCDs in lower-middle-income countries is changes in living environment development-associated urbanization [5]. For example, the consumption of fast foods high in sugar, fat and salt has become widespread, resulting in an increase in the prevalence of obesity and hypertension [6,7]. Further, physical activity has decreased due to improved transportation [8,9]. As a result, NCDs have been increasing more rapidly in urban areas than in rural areas. Despite ongoing rapid economic development in Cambodia, the medical care system has not fully caught up, even in urban areas, and necessary treatments for these diseases are not easily available. Therefore, the prevention of NCDs is an urgent issue.

Metabolic syndrome (MetS) is a combination of risk factors such as elevated blood pressure, dyslipidemia, hyperglycemia and obesity; it is a high-risk factor for NCDs [10].

The World Health Organization has identified unhealthy diets, physical inactivity, tobacco use and the harmful use of alcohol as lifestyle factors that contribute to the risk of NCDs [3]. Several previous studies have shown that these lifestyle factors influence the development of MetS [11,12,13,14,15,16]. Furthermore, some studies have clearly shown that lifestyle interventions, such as nutritional management and the promotion of exercise, reduces the risk of MetS [17,18,19,20]. In addition, eating habits, such as eating speed and skipping breakfast, walking speed and sleep, have also been reported to be associated with MetS [21,22,23,24]. Thus, in addition to diets and physical inactivity, various other lifestyle factors are relevant and warrant attention.

In 2010, the STEPS survey investigated lifestyle factors associated with the risk for NCDs in Cambodia [25]. It reported that smoking, alcohol consumption and low vegetable/fruit intake were risk factors for NCDs. Other cross-sectional studies have shown an association between fast food intake and obesity [26]. However, there is insufficient research on NCDs and lifestyles in Cambodia, with the existing data being reported tangentially as part of studies focused on different issues. Furthermore, there are no reports focusing on MetS in Cambodia, with such studies mainly originating from developed countries or China. Identifying the link between various lifestyle habits and MetS in Cambodia would allow for the development of multifaceted preventive measures, which may also be relevant to other low- and middle-income countries. Therefore, this study aimed to clarify the lifestyle habits associated with MetS in urban Cambodia, focusing not only on smoking, alcohol consumption and exercise, but also other lifestyle habits such as eating, walking speed and sleep.

## 2. Materials and Methods

### 2.1. Data Source and Study Population

In this cross-sectional study, we used data measured during a health checkup conducted at an international private general hospital, which is a collaborative research institution, with the permission of the collaborative research institution. This private hospital, which opened in 2016, is located in Phnom Penh, the capital of Cambodia, and aims to provide quality medical care equivalent to Japanese standards. The hospital targets the middle class to the affluent population and specializes in neurosurgery, gastrointestinal surgery and internal medicine. Most patients come from Phnom Penh or the surrounding area, which are urban areas [25]. In 2019, 39.4% of the total population of Cambodia, or approximately 6.14 million people, were living in urban areas [27].

The participants comprised Cambodians aged ≥20 years who underwent health checkups at this hospital between January 2017 and December 2019. During this period, 10,533 participants underwent health checkups at this hospital. Of these, those younger than 20 years (*n* = 89) and those who had two or more health checkups during the period were included in the analysis on their first visit, and duplicate data were excluded (*n* = 1675). Those with missing information to define MetS and lifestyle items necessary for analysis were also excluded (*n* = 3310). Finally, data from 5459 individuals were analyzed for this study (Figure 1).

### 2.2. Variables

The variables assessed were demographic charactersitics (age, sex), physical measurements, blood data and lifestyles and were obtained from electronic medical record data of the health checkups.

#### 2.2.1. Definition of Metabolic Syndrome

The harmonized diagnostic definition from the Joint Interim Statement was used to define MetS as a dependent variable [10]. MetS was diagnosed when three or more of the following five risk factors were met: (1) abdominal obesity (waist circumference [WC] ≥90 cm in men and ≥80 cm in women); (2) elevated triglycerides [TG] levels (≥150 mg/dL) or on treatment for dyslipidemia; (3) decreased high-density lipoprotein cholesterol [HDL-C] levels (<40 mg/dL in men, <50 mg/dL in women or ongoing treatment for dyslipidemia); (4) elevated blood pressure [BP] (≥130 mmHg systolic BP, ≥85 mmHg diastolic BP) or treatment for previously diagnosed hypertension; and (5) elevated fasting blood glucose [FBG] level (≥100 mg/dL) or treatment for previously diagnosed type 2 diabetes.

#### 2.2.2. Lifestyle Factors

The following 10 lifestyle factors are the independent variables. Three items related to exercise, “I do exercise lightly, sweating for at least 30 min per session more than twice a week for over a year”, “I walk or do equivalent physical activities in daily life for more than 1 h per day”, “My walking speed is faster than those of the same sex of similar age”; four items related to eating habits, “I eat quicker than others”, “I have dinner within 2 h before bedtime at least three days a week”, “I eat food after dinner at least three days a week”, “I skip breakfast at least three days a week”; an item related to sleep, “I get enough rest from sleep”; an item related to alcohol consumption, “I drink alcohol”; and an item related to smoking, “I currently smoke habitually”. Of the 10 items, the following seven items were answered with either “Yes” or “No”: “I do exercise lightly, sweating for at least 30 min per session more than twice a week for over a year”, “I walk or do equivalent physical activities in daily life for more than 1 h per day”, “My walking speed is faster than those of the same sex of similar age”, “I have dinner within 2 h before bedtime at least three days a week”, “I eat food after dinner at least three days a week”, “I skip breakfast at least three days a week” and “I get enough rest from sleep”. For the question, “I eat quicker than others”, the answers were “Quickly”, “Normal” and “Slowly”, and for “I drink alcohol” and “I currently smoke habitually”, the answers were “Current drinker/smoker”, “Former drinker/smoker” and “Never”.

#### 2.2.3. Anthropometric and Biochemical Measurements

The health checkups were conducted by medical professionals who had received specialized training in such assessments. All anthropometric indices were measured twice to reduce measurement error, and the average value was adopted. The data from electronic medical records comprised details such as systolic and diastolic BP, WC, body mass index [BMI: body weight {kg}/height {m}^2^]) and blood test results (TG, HDL-C, low-density lipoprotein cholesterol [LDL-C], FBG). Blood pressure was measured at the upper arm using a medical automatic blood pressure monitor (ES-H55, Terumo Corporation, Tokyo, Japan) after at least 5 min of rest. WC diameter is the diameter of the umbilical height measured while standing and during light exhalation. Blood tests were performed using an automated chemistry analyzer (Siemens Dimension EXL 200, Siemens Healthcare, Erlangen, Germany) at the hospital. Health checkup participants were required not to consume anything except water at least 8 h to avoid affecting the blood test results. Therefore, participants were instructed to finish dinner by 9:00 p.m. and avoid eating or consuming sweetened beverages thereafter before visiting the hospital, and health checkups were conducted in the morning.

### 2.3. Statistical Analysis

Analyses were performed separately by sex, and descriptive statistics of the subjects were presented. The data are presented as means and standard deviations for continuous variables and as numbers and percentages for categorical variables. The anthropometric and hematological values of the MetS and non-MetS groups were adjusted for age, and bivariate analysis was performed. To examine the association between MetS and lifestyle habits, multivariable logistic regression analysis was used, with the presence of metabolic syndrome as the dependent variable and lifestyle habits as the independent variable. First, univariate logistic regression analysis with age adjustment was performed to determine the variables of interest for the multivariable analysis (Model 1). Multivariable logistic regression analysis was conducted using the forced entry method with the variables for which *p* < 0.2 was obtained in the univariate logistic regression analysis as independent variables (Model 2). The results of each of the logistic regression analyses were presented with odds ratios and 95% confidence intervals [CI]. The Hosmer–Lemeshow goodness-of-fit test was used to evaluate the suitability of the final model.

IBM SPSS Statistics for Windows Ver.28 (IBM Corp., Armonk, NY, USA) was used for statistical analysis, and the significance level was set at 5% (two-tailed test).

### 2.4. Ethical Consideration

Ethical approval was obtained from the Cambodian National Ethics Committee for Health Research (200NECHR). Ethical approval was also obtained from the Collaborative Research Institution (No.20-002) and from the Ethics Review Committee of the Nagoya University School of Health Sciences (No. 20-109). The provided existing data were provided in fully anonymized form. Individual informed consent was waived by all Ethics Review Committees because the opt-out consent process was used in this study.

## 3. Results

### 3.1. Characteristics of Participants

A total of 5459 (2845 [52.1%] men, 2914 [47.9%] women) participants were included in the study. The prevalence of MetS was 60.4% in men and 52.6% in women. The statistical analysis for the characteristics of these participants is presented in Table 1.

### 3.2. Characteristics of Participants with and without MetS

Table 2 shows the physical characteristics and blood test results in the MetS and non-MetS groups. Compared to the non-MetS group, the MetS group showed significantly higher values for BMI, abdominal circumference, blood pressure, triglycerides, blood glucose and HbA1c and significantly lower values for HDL-C in both men and women (*p* < 0.001). Only women showed significantly higher LDL-C levels (*p* < 0.001).

### 3.3. Lifestyle Factors Associated with MetS

The results of the association between MetS and lifestyles by sex are shown in Table 3 and Table 4. Multivariable logistic regression analysis showed that the highest odds ratio for both men and women was “eating quicker than others”, with odds ratios of 2.25 (95% CI = 1.68–3.03, *p* < 0.001) and 1.92 (95% CI = 1.41–2.60, *p* < 0.001), respectively. Two other factors, “walking faster than those of the same sex of similar age “ (men: OR = 0.78, 95% CI = 0.67–0.92, *p* = 0.003, women: OR = 0.75, 95% CI = 0.62–0.89, *p* = 0.002) and “drinking alcohol” (men: OR = 1.33, 95% CI = 1.10–1.61, *p* = 0.004, women: OR = 1.33, 95% CI = 1.09–1.62, *p* = 0.004), showed significant associations with MetS. In addition, “eating speed is normal” (OR = 1.73, 95%CI = 1.30–2.31, *p* < 0.001) was significantly associated with MetS in men, while in women, “eating food after dinner at least 3 days a week” (OR = 1.25, 95% CI = 1.01–1.55, *p* = 0.041), “skipping breakfast at least 3 days a week” (OR = 0.83, 95% CI = 0.69–0.99, *p* = 0.045) and “getting enough rest from sleep” (OR = 1.19, 95% CI = 1.01–1.42, *p* = 0.043) were significantly associated with MetS.

## 4. Discussion

This study focused on the lifestyle habits of Cambodians living in urban Cambodia and is the first to show an association with MetS. The findings of this study indicated that the lifestyle factors significantly associated with MetS were walking faster than those of the same sex of similar age, eating quicker than others, eating food after dinner at least 3 days a week, skipping breakfast at least 3 days a week and getting enough rest through sleep. These related lifestyles were similar to the results obtained in previous studies in Western and developed countries, implying the possibility of the effectiveness of interventions through health education and health guidance for the prevention of MetS.

The prevalence of MetS was 56.6% (men: 60.4%, women: 52.6%), which comprised the majority and showed a high value. The study hospital was targeting the middle to the affluent class. Since most of the participants are urban residents, urbanization most likely affects their lifestyles. The high prevalence of MetS among the participants may also reflect urbanization. Furthermore, physical measurements and blood test data for both men and women indicated that those with MetS had poorer health than those without. These results suggest the necessity of lifestyle interventions for MetS.

The category with the highest odds ratio was “eating quicker than others” for both men and women. The association of eating speed with MetS has been shown in many previous studies, and the results were consistent in Cambodia. In a cohort study of Japanese participants, fast eaters had a higher risk of MetS than those who did not [28]. This study also revealed that fast eating was significantly associated with MetS components such as abdominal circumference and HDL-C. Studies in China and Korea have also reported similar findings [29,30]. The reasons why eating too fast is a risk for MetS include a delay in the brain’s perception of satiety, resulting in overeating and increased total energy intake [31,32], as well as its association with impaired insulin resistance, which is known to cause type 2 diabetes and visceral obesity [33]. Controlling the eating rate in Cambodia may reduce the occurrence of MetS, since these causes are physiological factors rather than variations across cultural backgrounds.

Alcohol consumption was associated with MetS, as in the previous study [30,34,35,36]. In this study, the answers were divided into three categories: current alcohol drinkers, quitters and never drinkers; however, many previous studies have focused on the frequency of drinking and changes in the amount and quantity of alcohol consumed [36,37,38]. These studies reported that those who drank more frequently, those who drank in greater quantities and those with increased alcohol consumption were more likely to have MetS. In contrast, some studies reported that light alcohol consumption reduces the risk of MetS [39]. Generally, alcohol consumption is considered to increase appetite, resulting in excessive energy intake and leading to obesity [40]. While this study did not examine the amount of alcohol consumed, the result suggests that excessive energy intake due to alcohol consumption may be associated with MetS.

Other significant associations were found only for women with “eating food after dinner at least 3 days a week”, “skipping breakfast at least 3 days a week” and “getting enough rest with sleep”, suggesting that women need to be more careful about their eating habits than men to avoid developing MetS and comorbid conditions.

According to previous studies, skipping breakfast is a risk factor for MetS and its components [41,42,43]. This is because skipping breakfast causes people to overeat on an empty stomach, leading to blood sugar spikes, in turn causing abdominal obesity and increased cholesterol [44]. Conversely, missing breakfast tended to lower the risk of MetS in a South Korean study, consistent with our results [45]. Other studies have reported that the group with missed breakfast had lower total energy intake [46,47]. The results of this study may also be attributed simply to a reduction in total energy intake due to the absence of breakfast. Similarly, the significant result of eating an evening meal after dinner as a risk for MetS may also be related to the increased total energy intake. Eating foods other than the main meal has been noted to lead to excess energy intake [48,49], and it is considered that energy tends to be stored more if consumed after dinner since there is less opportunity for expending energy.

Regarding the association of MetS with sleep, sleep duration and sleep quality reportedly affect MetS. Oversleeping, insufficient sleep and poor-quality sleep are reportedly associated with the risk of MetS [50,51,52,53]. Herein, we did not investigate sleep duration or quality but only asked about subjective assessments. Therefore, there is not enough information to interpret why there was a positive association between women who felt well rested through sleep and MetS. Nevertheless, the present results are interesting and should be considered for the duration and quality of sleep among Cambodians.

In this study, there was no association between exercise and MetS. However, “faster walking speed compared to the same sex at the same age” was negatively associated with MetS in both men and women. Walking speed is sometimes used to indicate exercise habits and physical capability. People who watched TV for shorter hours reportedly had faster walking speeds than those who watched TV for longer hours [54] and that walking speed can be used as an indicator of physical frailty and physical independence [55,56]. This is because walking speed correlates with both physical activity and exercise tolerance. The present results also suggest that walking speed reflects physical activity and exercise tolerance and thus could be a low-risk factor for MetS.

This study has several limitations. First, a causal relationship between MetS and lifestyle cannot be stated because this is a cross-sectional study. Second, the results may not be generalizable as the participants were from a single institution. Furthermore, there is a possibility of sampling bias, such as lifestyle bias, due to the study hospital targeted the middle class to affluent population. Fourth, there is a risk of cognitive bias in the answers to the lifestyle questions since they are the subjective responses of the participants. In addition, there is a risk of observer bias, such as measurement error among measurers, as the data were not specifically obtained for research purposes. Another limitation is that the Hosmer–Lemeshow *p*-value for men was 0.007, which was a poor fit of the model, and there may have been other factors influencing the results. Finally, we were able to clarify the relationship between previous studies’ lifestyle habits and MetS in Cambodia because the present study used existing health examination questionnaire items, but we may not have been able to derive sufficient information on lifestyle factors unique to Cambodia.

## 5. Conclusions

Our results indicated that lifestyles significantly associated with MetS among Cambodians living in urban areas of Cambodia were eating quicker than others, walking faster than those of the same sex of similar age and drinking alcohol in both men and women. In addition, there were significant associations with normal eating speed for men only and, for women only, skipping breakfast at least three days per week, eating food after dinner at least three days per week and getting enough rest through sleep. Further studies are needed to examine lifestyle factors to derive more suitable content for Cambodians’ living environment, which may lead to efforts for the prevention of MetS tailored to the characteristics of Cambodians.

## Figures and Tables

**Figure 1 ijerph-19-10481-f001:**
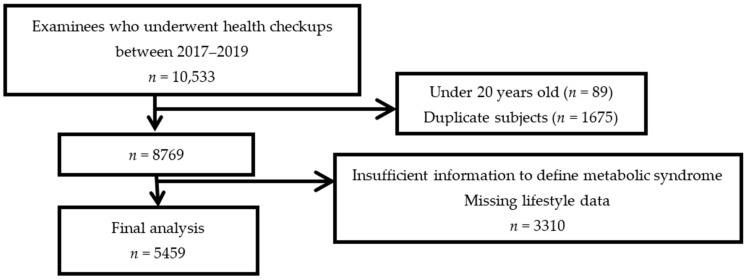
Flowchart of the selection of the study population.

**Table 1 ijerph-19-10481-t001:** Characteristics of participants by sex.

	Men	Women
Number (%)	2845 (52.1)	2614 (47.9)
Age (years)	47.0 ± 14.4	49.4 ± 15.6
MetS *n* (%)	1717 (60.4)	1375 (52.6)
BMI	25.5 ± 3.6	23.9 ± 4.0
WC (cm)	88.7 ± 10.0	80.5 ± 10.7
Systolic BP (mmHg)	128.6 ± 16.4	122.2 ± 19.0
Diastolic BP (mmHg)	84.5 ± 11.1	77.8 ± 11.3
Triglycerides (mg/dL)	186.45 ± 150.23	135.13 ± 101.1
FBG (mg/dL)	110.0 ± 31.7	103.6 ± 27.0
HbA1C (%)	5.8 ± 0.98	5.8 ± 0.95
HDL-C (mg/dL)	41.9 ± 10.8	49.2 ± 13.0
LDL-C (mg/dL)	122.8 ± 34.4	125.3 ± 36.0

Mean ± standard deviation. Abbreviation: BMI; body mass index, WC; waist circumference, BP; blood pressure, FBG; fasting blood glucose, HbA1C; hemoglobin A1C, HDL-C; high-density lipoprotein cholesterol, LDL-C; low-density lipoprotein cholesterol.

**Table 2 ijerph-19-10481-t002:** Characteristics of participants with MetS and non-MetS.

	Men (*n* = 2845)			Women (*n* = 2614)		
	MetS	Non-MetS	*p*-Value	MetS	Non-MetS	*p*-Value
Number (%)	1717 (60.4)	1128 (39.6)		1375 (52.6)	1239 (47.4)	
Age (years)	49.4 ±13.7	43.3 ± 14.5	<0.001 ^a^	55.6 ± 14.0	42.5 ± 14.4	<0.001 ^a^
BMI	26.7 ± 3.4	23.6 ± 3.0	<0.001 ^b^	25.6 ± 3.7	22.0 ± 3.3	<0.001 ^b^
WC (cm)	92.7 ± 9.0	82.7 ± 8.4	<0.001 ^b^	85.8 ± 8.9	74.5 ± 9.2	<0.001 ^b^
Systolic BP (mmHg)	133 ± 16.4	122.3 ± 14.3	<0.001 ^b^	129.8 ± 18.6	113.7 ± 15.7	<0.001 ^b^
Diastolic BP (mmHg)	87.3 ± 11.1	80.4 ± 14.3	<0.001 ^b^	81.5 ± 11.2	73.8 ± 12.3	<0.001 ^b^
Triglycerides (mg/dL)	232.2 ± 172.3	116.9 ± 61.0	<0.001 ^b^	175.7 ± 115.9	90.1 ± 52.9	<0.001 ^b^
FBG (mg/dL)	116.7 ± 36.8	99.9 ± 16.7	<0.001 ^b^	112.0 ± 33.4	94.3 ± 11.7	<0.001 ^b^
HbA1c (%)	6.0 ± 1.1	5.5 ± 0.7	<0.001 ^b^	6.1 ± 1.1	5.4 ± 0.5	<0.001 ^b^
HDL-C (mg/dL)	38.4 ± 9.2	47.2 ± 10.8	<0.001 ^b^	43.6 ± 10.9	55.5 ± 12.3	<0.001 ^b^
LDL-C (mg/dL)	129.4 ± 37.1	127.8 ± 32.7	0.061 ^b^	122.9 ± 37.3	120.1 ± 33.1	<0.001 ^b^

Mean ± standard deviation. Abbreviation: BMI; body mass index, WC; waist circumference, BP; blood pressure, FBG; fasting blood glucose, HbA1C; hemoglobin A1C, HDL-C; high-density lipoprotein cholesterol, LDL-C; low-density lipoprotein cholesterol. ^a^: Student’s *t*-test, ^b^: age-adjusted linear regression analysis.

**Table 3 ijerph-19-10481-t003:** Association between metabolic syndrome and lifestyle (men).

Lifestyle Items (10 Items) Men	Number (%)		Age-Adjusted Logistic Regression Model 1 OR (95% CI)	*p*-Value	Multivariable Logistic Regression Model 2 OR (95% CI)	*p*-Value
	MetS (*n* = 1717)	Non-MetS (*n* = 1128)				
Exercising with light sweat for at least 30 min per session more than twice a week for over a year						
Yes	1023 (59.6)	664 (58.9)	0.92 (0.79–1.08)	0.324	-	
No	694 (40.4)	464 (41.1)	1			
Walking or equivalent physical activity in daily life for more than 1 h per day						
Yes	945 (55.0)	619 (54.9)	0.95 (0.81–1.11)	0.528	-	
No	772 (45.0)	509 (45.1)	1			
Walking faster than the same sex of similar age						
Yes	718 (41.8)	526 (46.6)	0.86 (0.74–1.00)	0.053	0.78 (0.67–0.92)	0.003
No	999 (58.2)	602 (53.4)	1		1	
Eating quicker than others						
Quickly	646 (37.6)	351 (31.1)	2.23 (1.66–2.99)	<0.001	2.25 (1.68–3.03)	<0.001
Normal	953 (55.5)	655 (58.1)	1.74 (1.32–2.31)	<0.001	1.73 (1.30–2.31)	<0.001
Slowly	118 (6.9)	122 (10.8)	1		1	
Having dinner within 2 h before bedtime at least 3 days a week						
Yes	909 (52.9)	562 (49.8)	1.13 (0.97–1.32)	0.105	1.12 (0.95–1.32)	0.175
No	808 (47.1)	566 (50.2)	1		1	
Eating food after dinner ≥3 days a week						
Yes	382 (22.2)	219 (19.4)	1.23 (1.01–1.48)	0.036	1.19 (0.98–1.46)	0.086
No	1335 (77.8)	909 (80.6)	1		1	
Skipping breakfast ≥3 days a week						
Yes	576 (33.5)	385 (34.1)	1 (0.85–1.18)	0.974		
No	1141 (66.5)	743 (65.9)	1			
Sufficient rest with sleep						
Yes	1069 (62.3)	686 (60.8)	1.05 (0.90–1.23)	0.53		
No	648 (37.7)	442 (39.2)	1			
Alcohol drinking habit						
Current drinker	1241 (72.3)	811 (71.9)	1.34 (1.11–1.62)	0.002	1.33 (1.10–1.61)	0.004
Former drinker	90 (5.2)	49 (4.3)	1.04 (0.70–1.54)	0.854	1.03 (0.69–1.53)	0.895
Never	386 (22.5)	268 (23.8)	1		1	
Smoking status						
Current smoker	169 (9.8)	94 (8.3)	1.13 (0.86–1.48)	0.388		
Former smoker	236 (13.8)	102 (9.1)	1.02 (0.78–1.34)	0.86		
Never	1312 (76.4)	932 (82.6)	1			

OR: odds ratio; 95% CI: 95% confidence interval. Model 1: adjusted for age, Model 2: adjusted for age and variables; *p* < 0.2 (Model 1). Model 2: Hosmer–Lemeshow *p* = 0.007.

**Table 4 ijerph-19-10481-t004:** Association between metabolic syndrome and lifestyle (women).

Lifestyle Items (10 Items) Women	Number (%)		Age-Adjusted Logistic Regression Model 1 OR (95% CI)	*p*-Value	Multivariable Logistic Regression Model 2 OR (95% CI)	*p*-Value
	MetS (*n* = 1375)	Non-MetS (*n* = 1239)				
Exercising with light sweat for at least 30 min per session more than twice a week for over a year						
Yes	683 (49.7)	543 (43.8)	0.99 (0.81–1.15)	0.694	-	
No	692 (50.3)	696 (56.2)	1			
Walking or equivalent physical activity in daily life for more than 1 h per day						
Yes	755 (54.9)	650 (52.5)	0.92 (0.77–1.09)	0.312	-	
No	620 (45.1)	589 (47.5)	1			
Walking faster than the same sex of similar age						
Yes	448 (32.6)	504 (40.7)	0.81 (0.68–0.97)	0.022	0.75 (0.62–0.89)	0.002
No	927 (67.4)	735 (59.3)	1		1	
Eating quicker than others						
Quickly	401 (29.2)	292 (23.6)	1.82 (1.34–2.46)	<0.001	1.92 (1.41–2.60)	<0.001
Normal	820 (59.6)	795 (64.2)	1.31 (0.99–1.73)	0.056	1.28 (0.97–1.28)	0.082
Slowly	154 (11.2)	152 (12.2)	1		1	
Having dinner within 2 h before bedtime at least 3 days a week						
Yes	706 (51.3)	645 (52.1)	0.96 (0.81–1.13)	0.591		
No	669 (48.7)	594 (47.9)	1			
Eating food after dinner ≥3 days a week						
Yes	302 (22.0)	219 (19.2)	1.24 (1.01–1.53)	0.043	1.25 (1.01–1.55)	0.041
No	1073 (78.0)	909 (80.8)	1		1	
Skipping breakfast ≥3 days a week						
Yes	424 (30.8)	436 (35.2)	0.84 (0.70–1.00)	0.053	0.83 (0.69–0.99)	0.045
No	951 (69.2)	803 (64.8)	1		1	
Sufficient rest with sleep						
Yes	811 (59.0)	691 (55.8)	1.19 (1.01–1.42)	0.043	1.19 (1.01–1.42)	0.043
No	564 (41.0)	548 (44.2)	1		1	
Alcohol drinking habit						
Current drinker	355 (25.8)	414 (33.4)	1.33 (1.09–1.61)	0.004	1.33 (1.09–1.62)	0.004
Former drinker	19 (1.4)	7 (0.6)	1.96 (0.78–4.97)	0.155	1.95 (0.77–4.94)	0.159
Never	1001 (72.8)	818 (66.0)	1		1	
Smoking status						
Current smoker	7 (0.5)	2 (0.2)	2.97 (0.56–15.91)	0.203	-	
Former smoker	5 (0.4)	2 (0.2)	1.17 (0.18–7.54)	0.87		
Never	1363 (99.1)	1235 (99.6)	1			

OR: odds ratio; 95% CI: 95% confidence interval. Model 1: adjusted for age, Model 2: adjusted for age and variables; *p* < 0.2 (Model 1). Model 2: Hosmer–Lemeshow *p* = 0.059.

## Data Availability

The datasets generated during and/or analyzed during the current study are available from the corresponding author on reasonable request.

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
