# Peer review of "Lifestyle Factors Associated with Metabolic Syndrome in Urban Cambodia"

_ijerph, 2022, doi:10.3390/ijerph191710481_

Round 1

Reviewer 1 Report

This cross-sectional study aimed to identify lifestyle factors associated with metabolic syndrome (MetS) in urban Cambodia. Overall, this is a well-written and interesting study; however, there are some areas of concern that should be considered:

Introduction: Please expand why lifestyle factors, such as diet and exercise habits could impact MetS. The authors need to highlight the novelty of this investigation at the literature and the difference between others.

Methods: Lifestyle factors (Line 112-130): Is there a specific reason to include these factors? Please, insert a reference or reason for that.

Anthropometric Measurements and Biochemical Measurement (Line 132-142): I understand that authors obtained data from electronic medical record, however, it would be important to know the Intra-class Correlation (ICC), since different people should be conducted the assessment at the hospital and the error between evaluators could influence the assessment and classification.

Discussion: Line 221-224: The authors said: “not missing breakfast” but they used in the methods and results “Skipping breakfast ≧ 3 days a week”. The same for “Walking faster than the same sex at the same generation” but they used in the discussion section “slow walking speed”. Please, it’s important to standard if not confused the readers, since the questions used previous were different. Please, also check it in the conclusion section.

Line 228: Please, change “Met S” to “MetS”.

Limitation: please, to insert the risk of bias about the classification of lifestyle, since the authors used subjective assessments.

Reviewer 2 Report

This study is considered to be meaningful as a study to clarify the relationship between metabolic syndrome and lifestyle habits in that the health effects of NCDs are also increasing in Cambodia, and metabolic syndrome is an important risk factor for cardiovascular disease or diabetes among NCDs..

However, there are some points which they should reconsider as follows.

1. The authors identified 10 items as lifestyle-related variables. Please provide evidence that these variables (questions) are appropriate tools to measure physical activity, dietary intake, sleep, smoking and alcohol consumption.

2. A multivariate analysis was performed on the relationship between metabolic syndrome and lifestyle, and it seems necessary to confirm whether this model is suitable.

3. The subjects of this study are people living in urban areas of Cambodia, and there may be differences from the overall population, so socioeconomic variables should be presented in the characteristics of the subjects and their control should be considered in the analysis.

4. In line 104, five symptoms doesn't seem to be the right word.

5. In line 135, it is likely that the amount of time the subjects fasted before the blood test was given.

6. Table 3 shows the proportion of metabolic syndrome and non-metabolic syndrome for each variable. Please present N as well.

This study is considered to be meaningful as a study to clarify the relationship between metabolic syndrome and lifestyle habits in that the health effects of NCDs are also increasing in Cambodia, and metabolic syndrome is an important risk factor for cardiovascular disease or diabetes among NCDs..

However, there are some points which they should reconsider as follows.

1. The authors identified 10 items as lifestyle-related variables. Please provide evidence that these variables (questions) are appropriate tools to measure physical activity, dietary intake, sleep, smoking and alcohol consumption.

2. A multivariate analysis was performed on the relationship between metabolic syndrome and lifestyle, and it seems necessary to confirm whether this model is suitable.

3. The subjects of this study are people living in urban areas of Cambodia, and there may be differences from the overall population, so socioeconomic variables should be presented in the characteristics of the subjects and their control should be considered in the analysis.

4. In line 104, five symptoms doesn't seem to be the right word.

5. In line 135, it is likely that the amount of time the subjects fasted before the blood test was given.

6. Table 3 shows the proportion of metabolic syndrome and non-metabolic syndrome for each variable. Please present N as well.

Round 2

Reviewer 2 Report

Thank the authors for their sincere responses to the review comments.

However, there are already some points which they should reconsider as follows.

1. As authors mentioned, The Hosmer-Lemeshow test is a goodness of fit test for logistic regression, especially for risk prediction models. A goodness of fit test tells you how well your data fits the model. So, maybe authors would be good to provide the result of it. In addition, it would be additionally necessary to demonstrate the explanatory power of models that can show the extent to which lifestyle habits influence metabolic syndrome.

2. As you explain, if you write the consideration of fasting time in this study in the text, it will help the reader to understand the general post-fasting test.
